# Neurotropic Viruses as Acute and Insidious Drivers of Aging

**DOI:** 10.3390/biom15040514

**Published:** 2025-04-01

**Authors:** Angela Rocchi, Hassen S. Wollebo, Kamel Khalili

**Affiliations:** Center for Neurovirology and Gene Editing, Department of Microbiology, Immunology and Inflammation, Lewis Katz School of Medicine, Temple University, Philadelphia, PA 19140, USA; rocchi@temple.edu

**Keywords:** neurotropic viruses, aging, neurodegeneration, dementia, proteinopathy, genome instability, senescence, herpesviruses, polyomaviruses, coronaviruses

## Abstract

Aging is the result of various compounding stresses that gradually overcome the homeostatic regulation of the cell, resulting in irreversible damage. This manifests as many acute and chronic conditions, the most common of which are neurodegeneration and dementia. Epidemiological studies have shown significant, strong correlations between viral infection and neurodegenerative diseases. This review overlays the characteristics of viral pathogenesis with the hallmarks of aging to discuss how active and latent viruses contribute to aging. Through our contextualization of myriad basic science papers, we offer explanations for premature aging via viral induction of common stress response pathways. Viruses induce many stresses: dysregulated homeostasis by exogenous viral proteins and overwhelmed protein quality control mechanisms, DNA damage through direct integration and epigenetic manipulation, immune-mediated oxidative stress and immune exhaustion, and general energy theft that is amplified in an aging system. Overall, this highlights the long-term importance of vaccines and antivirals in addition to their acute benefits.

## 1. Introduction

“Aging” describes the gradual damage cells acquire over time, culminating in permanently impaired function and, ultimately, death [1]. In the central nervous system, neurotropic viruses are a major stressor and, therefore, a major driver of aging. Epidemiological studies of multiple large biobanks have shown that patients with a history of neurological viral infection are thirty times more likely to develop a neurodegenerative disease [2]. In addition to the pressures of viral proteins during active infection, acute and chronic viral infections disrupt the homeostasis of the cell [3]. This occurs throughout life, with acute causes of neurodegeneration (e.g., hemorrhagic or ischemic stroke, traumatic brain injury) and chronic conditions (e.g., viral latency, metabolic disease) compounding, often synergistically [4]. Here, we define aging, outline the impact viruses have on the brain, and identify the overlapping pathways of viral pathogenesis and age-related neurodegeneration. Previously proposed “Hallmarks of Aging” range in number but can be generally described by three categories: altered proteostasis, genomic compromise, and senescence (Figure 1) [1]. Neurotropic viruses manipulate each of these categories, driving rapid neurodegenerative diseases like Amyotrophic Lateral Sclerosis (ALS) and Parkinson’s Disease (PD), and more progressive neurodegenerative conditions like Alzheimer’s Disease (AD) and Frontotemporal Dementia (FTD) [2]. In fact, the neuro-penetrance of these viruses is so extensive that they have been widely proposed as a non-invasive vector for therapeutic delivery to the brain [5,6]. Specific to aging, viral vectors have been proposed to deliver neurotrophic factors for the treatment of PD [7]; genetic interventions post-ischemic stroke [8,9]; GABA antagonists for the treatment of cognitive decline [10]; and epigenetic modulators for the intervention of epigenetic aging [11].

Although not the sole cause of aging, this review highlights the many ways in which insidious viral infections extend beyond transient colds into lasting, premature aging of the brain [12]. It is important to acknowledge that aging throughout the body has implications for the brain [13,14,15,16,17]. However, this review focuses on the direct stressors enacted by non-fatal, neurotropic viruses (Table 1). This includes linear positive sense ssRNA coronaviruses, like novel coronavirus SARS-CoV-2; enteroviruses coxsackievirus (CV), echovirus, and poliovirus; flaviviruses dengue, Japanese encephalitis virus (JEV), and West Nile virus (WNV); retrovirus human immunodeficiency virus (HIV); linear dsDNA herpesviruses herpes simplex 1 (HSV1), varicella zoster (VSV), Epstein–Barr (EBV), and cytomegalovirus (CMV); and circular dsDNA JC polyomavirus (JCV).

## 2. Altered Proteostasis

Protein quality control (PQC) describes the interconnected pathways of a cell that identify misfolded proteins and direct (1) refolding in ideal conditions, (2) degradation by proteolysis/autophagy for molecular recycling, or (3) controlled apoptosis in high-stress, low-nutrient states [19]. Due to its regulatory role, PQC is interconnected with nutrient sensing, mitochondrial stability, and cell cycle regulation. The most common age-related brain disorders are proteinopathies: tau and amyloid-β (Aβ) aggregation define AD, α-synuclein aggregates characterize both PD and Lewy body dementia (LBD), and TDP-43 aggregates drive pathogenesis of ALD and FTD [20,21,22]. Evidence discussed here indicates that viral infection not only drives aggregation of proteins but also manipulates the PQC capacity of a cell, preventing the cell from reinstituting proteostasis (Table 2).

### 2.1. Viral Aggregation of Intracellular Tau and Extracellular Amyloid-β

Extracellular Aβ plaques and intracellular tau tangles are pathognomonic for AD [20]. In a healthy system, both Aβ and tau function as antiviral mediators [51,52]. Prior to entering the host cell, viral capsid and envelope proteins accrue an obstructive “corona” of host proteins, including Aβ, trapping the virus in the extracellular matrix (Figure 2a). This physical obstruction prevents capsid interactions with host cell membranes, impairing cell entry (Figure 2b). This antimicrobial role of Aβ is supported by an increased susceptibility to infection observed following pharmacologic reduction of Aβ [53]. In excess, however, the physical accumulation of antiviral Aβ on compatible viral proteins becomes a nidus for amyloid aggregation (Figure 2c) [54,55]; HSV1 gD, for example, is an envelope glycoprotein that directly agglutinates Aβ via intermolecular hydrogen bonding and van der Waals forces [26]. The heparin binding region of S-protein, the eponymous spike protein of SARS-CoV-2, is exceptionally aggregation-prone, physically binding aggregates of Aβ in the extracellular matrix as well as tau, TDP43, and α-synuclein within the cytoplasm [27]. In fact, S-protein has such exceptional proficiency for amyloidosis, it has been shown to produce toxic aggregates within the vasculature and other organs, as well as brain tissue.

Another herpesvirus, CMV, utilizes this proteinopathy to its advantage, with viral protein M45 independently assembling into amyloid fibrils, impairing host access to virions and preventing virus-induced necroptosis of the host [23]. Similarly, HIV TAT—a secreted viral transcription factor—not only binds extracellular Aβ, it also synergizes with it, increasing neurotoxicity compared to Aβ alone [24,25]. 

When the virus achieves cell entry, host cell processing of viral DNA and proteins activates the cGAS-STING pathway, an innate antiviral response (Figure 2d) [56,57]. cGAS directly identifies abnormal DNA or proteins in the cytoplasm, triggering STING to activate NFκB and IRF3 (Figure 2e). Downstream phosphorylation of NFκB, IRF3, and tau by TBK1 enables cGAS-STING-dependent degradation of viral proteins and cytokine activation of the immune response. This was directly evidenced with the ICP27 protein of HSV1, which triggered cGAS-STING antiviral activity as well as tauopathy in neurons and microglia of AD patients [32]. Independent of HSV1, activation of cGAS-STING alone is sufficient to induce tau-phosphorylation, improving neuron survivability. This suggests that phosphorylation of tau is an antiviral response requisite for the prevention of acute neurodegeneration. In an abnormal PQC state, however, chronic hyperphosphorylation of tau leads to aggregation, proteinopathy, and neurodegeneration (Figure 2f). The viral nucleic acids of CMV, HIV, and SARS-CoV-2 also trigger the cGAS pathway [34,58,59]. The antiviral effect of the cGAS is underscored by the evolution of anti-cGAS viral proteins that directly or indirectly impair its triggering of the cascade (Figure 2g). This includes CMV tegument proteins pUL31 and pUL83; Dengue NS2B, NS3, NS2B3; HIV capsid protein GAG; HSV1 tegument protein VP22; and VZV tegument protein ORF9 [28,29,30,31,33,35,60,61].

### 2.2. Viral Aggregation of TDP43

TDP43 aggregates are implicated in motor neuron degeneration in ALS, as well as cortex neuron degeneration in FTD [22]. Pro-aggregatory genetic mutants of TDP43 are implicated in familial ALS and FTD, but aggregates of wildtype TDP43 have also been shown to form following viral infection, including neurotropic HIV, HERV-K, and SARS-Cov-2 [27,36,41]. The most established TDP43 translocating viruses are the positive-sense RNA enteroviruses, including coxsackievirus, echovirus, and poliovirus [37,38,39]. Normally, transactive response DNA-binding protein 43 is a general regulator of host RNA processing in the nucleus [62]. During infection, TDP43 interacts with viral RNA, resulting in translocation to the cytoplasm where toxic aggregates form (Figure 2j,l) [63]. As with hyperphosphorylated tau tangles and production of Aβ fibrils, the translocation of TDP43 has been posited as an antiviral response, with aggregates obstructing viral entry into the nucleus (Figure 2k) [40]. Not only does this cytoplasmic aggregation drive specific neurodegenerative conditions ALS and FTD, but evidence also suggests TDP43 translocation can impair general memory through its proteinopathy [64]. This disruption produces an IFN response independent from cGAS-STING activation, indicating a separate proinflammatory proteinopathy from the intracellular tauopathies discussed.

### 2.3. Viral Aggregation of α-Synuclein

Aggregation of α-synuclein is the root proteinopathy of both LBD and PD [65]. Whereas LBD exhibits diffuse cytotoxic “Lewy Bodies” within cortical neurons and first present with cognitive impairments, Lewy body localization within dopaminergic neurons of the substantia nigra presents with the rigid motor dysfunction characteristic of PD [21]. In a healthy system, α-synuclein functions to traffic synaptic vesicles for neurotransmitter release. As with Aβ, tau, and TDP43, evidence suggests α-synuclein also has an antiviral function: redirecting virion-containing endosomes to the endoplasmic reticulum (ER) (Figure 2h). In addition to genetic predisposition, viral infection has been shown to increase α-synuclein expression and aggregation (Figure 2i) [48]. Knockout models of α-synuclein produced a five-fold increase in the viral load of West Nile virus encephalitis, whereas a healthy model exhibited a virus-stimulated increase in α-synuclein expression and accompanying sequestration of virions to the endoplasmic reticulum in a Rab1-dependent manner [44]. In addition to impairing the production of viral progeny, this enables α-synuclein modification of the PQC mechanism of the cell, preventing premature triggering of the caspase-3-dependent intrinsic apoptotic cascade. Because the antiviral role of α-synuclein is unique to neurons, its function—and dysfunction—has been correlated with additional neurotropic viruses, including other flaviviruses (i.e., dengue, JEV), herpesviruses (CMV, EBV), HIV, and SARS-CoV-2 [42,45,48,49,50]. Concordantly, treatment with antivirals has been correlated clinically with a reduced risk of Parkinsonism [66].

### 2.4. Viral Aberration of Protein Quality Control

Presuming the tau, amyloid, TDP43, and α-synuclein responses defend against viruses under normal conditions, disruption of the PQC regulatory system must occur to shift the system from protective to pathogenic. In identifying foreign proteins, PQC redirects viral peptides toward degradation by proteolysis or autophagy [67,68]. In the former mechanism, basal ubiquitination of substrates directs aberrant or foreign proteins to be degraded in the proteosome complex; amino acids are then recycled into other polypeptides [69]. In the latter mechanism, a stress-induced chaperone identifies a substrate, engulfs it in a lysosome, and directs its lysosomal degradation through acidic pH and proteases [70]. CMV, EBV, HIV, HSV1, and SARS-CoV-2 express deubiquitinases that remove ubiquitin from proteins the host cell has marked for degradation by lysosomes or proteosomes [71]. In instances where lysosomes or proteosomes do reach viral proteins, “hijacking” of PQC chaperones is a characteristic of many viruses. Neurotropic coronaviruses, enteroviruses, and polyomaviruses have all been shown to redirect lysosomes from their traditional autophagic pathways to deliver virions to the nucleus and viral progeny to the cell membrane [72,73,74]. Coronaviruses have also been shown to raise lysosome pH, impairing enzyme function, preventing degradation following endocytosis-mediated cell entry [75].

When stresses exceed the nutrient source of a cell, healthy PQC enactors shift the “cell fate” toward apoptosis [67]. By eliminating the host machinery necessary for viral gene and protein production, this pro-apoptotic shift is inherently antiviral [76]. It is therefore an evolutionary advantage for viruses to immortalize cells, prolonging access to the transcriptional and translational machinery on which they depend [77]. JCV, for example, inhibits tumor suppressors p53 and pRb through the eponymous transcription factor “large tumor antigen” [78]. This is in direct conflict with the endogenous PQC pathway, which directs cell fate toward apoptosis during the increased stress of active infection [76]. Other anti-apoptotic manipulations of PQC include CMV inhibition of pro-apoptotic JNK via viral protein, pUL38; EBV sequestration of stress indicator HSP70 via viral protein, EBNA-LP; HSV1 bypassing of nutrient sensing to ensure protein expression via viral proteins gB and γ34.5; and endogenous retrovirus stimulation of stemness via OCT4 expression [77,79,80,81]. These manipulations allow metabolic stress to exceed the pre-determined maximum set by PQC, further advancing aging [82].

In this section, we discussed how endogenous proteins—such as tau, Aβ, TDP43, and α-synuclein—react to viral infection and impair viral pathogenesis. It is essential to reiterate that an intact PQC system can suppress acute infection and return the cell to proteostasis. In the cases discussed—chronic viral infection, pro-aggregatory antiviral mediators, or disrupted PQC—stressors can compound, resulting in permanent proteinopathy and irreversible damage. The extremes of these conditions present as common neurodegenerative diseases (e.g., AD, ALS, FTD, PD), but their subclinical or undiagnosed effects produce strain conducive to premature aging. In “microdosing” these pro-degenerative stressors, dysregulation of PQC enables gradual progression toward neurodegeneration.

## 3. Genomic Compromise

Similar to PQC, the DNA damage response (DDR) relies on myriad sensor and enactor molecules to identify and repair strand breaks and aberrations [83]. At checkpoints of the cell cycle, recognized damage is addressed prior to advancement. At the G1S junction, p53, pRb, and p21 tumor suppressors are most active in identifying aberrant genes, pausing progression to DNA synthesis until genome stability has been established (Figure 3a–c). At the G2M junction, ATM and ATR identify double and single-stranded breaks, respectively [84,85]. ATM activates p53 and Chk, ATR activates Chk2, and altogether these DDR enactors halt cell cycle progression to mitosis (Figure 3g–i). For cells to replicate or function during genomic compromise, the DDR must succeed at repairs or be overridden in spite of damage. Irreversible and/or insurmountable genome compromise is known to increase with natural aging [1]. Viruses contribute to this stress, and by extension, promote cell death or senescence; it becomes essential that viruses evolve countermeasures to the DDR in order to maintain a long-term host and source of protein machinery [86]. In this section, we discuss how age-associated genomic compromise—DNA damage, telomere attrition, and epigenetic shift—are generated by viral infection.

### 3.1. Viral Damage to Host DNA

The most overt clastogenic damage occurs during integration. HIV is perhaps the most defined integrating lentivirus; HIV encodes an integrase protein that cleaves the host DNA, inserts the viral DNA, and directs host DNA repair pathways to seal the exogenous DNA in place rather than recognizing it as foreign [87]. This is critical to HIV pathogenesis, with integrase inhibitors (e.g., cabotegravir) utilized for both prophylactic and therapeutic highly active antiretroviral therapy regimens. In the absence of HAART, however, HIV predominantly integrates into introns of active genes [88]. Similar to its preferential infection of CD4 lymphocytes in the periphery, HIV integrates into microglia of the brain. In HIV neurocognitive disorders, integration into astrocytes is also reported [89]. Although HIV integrates semi-randomly, BACH2 and MKL2 integration sites are known to be pro-proliferative and, therefore, selectively persist following infection [90]. The resultant increase in microglia activation advances aging twofold: through rapid increases in proliferative and inflammatory stress [91].

Integration is critical to retroviruses, such as HIV, but is also reported for dsDNA herpesviruses EBV and CMV [92]. EBV—the causative agent of mononucleosis, multiple sclerosis, and multiple cancers, including nasopharyngeal carcinoma, B-cell leukemias, and Burkitt lymphoma—is known to integrate at “fragile points” of the host genome [93,94]. This includes the telomeric repeats (see Section 3.2 for functional description), resulting in TERT-dependent upregulation of the cell cycle. Association of viral protein EBNA1 with the host chromosome can induce complete breakage and rearrangement of the host chromosome at site 11q23 [95]. The strain this causes to the host does not stop there. In a healthy system, these double-stranded DNA breaks are identified by the ATM protein at the G2M checkpoint (Figure 3g). ATM (and single strand break counterpart ATR) is responsible for recruiting enactors of the DNA repair response (Figure 3h). These proteins activate Chk1/2 to pause the cell cycle until repair needs have been met (Figure 3i).

To avoid recognition as foreign and malignant DNA, viruses promote ATM activity (ensuring DNA repair molecules are recruited to facilitate their integration) while impairing Chk1/2 (ensuring progression through the cell cycle) (Figure 3j,k). There are myriad mechanisms by which viruses achieve this (Table 3). The VPR protein of HIV, for example, activates ATR through direct chromatin binding while simultaneously inactivating Chk1 through phosphorylation [96,97]. Conversely, SARS-CoV-2 produces two separate viral proteins, ORF6 and NSP13, which both drive degradation of Chk1 through the separate pathways of proteolysis and autophagy, respectively [98].

In addition to the direct oxidative stress caused by this DNA manipulation, less drastic integration sites can produce more gradual stress through increased proliferative strain [101]. DNA damage as a hallmark of aging is significantly amplified by viral modifications of the DNA damage response, enabling aberrant DNA to persist and accumulate stress. The tumor suppressor p53 is the paramount regulator of the cell cycle [103]. Most expressed during the G1S junction, p53 activates in response to foreign or abnormal DNA (Figure 3a). It upregulates transcription of the tumor suppressor, p21, and triggers disinhibition of the tumor suppressor pRb. By halting the cell cycle and directing either DDR or apoptosis, p53 and its downstream mediators prevent persistence or exacerbation of gene instability (Figure 3b). Next, during S phase, p53 promotes telomerase elongation (discussed in detail in Section 3.2), ostensibly in anticipation of the stress produced by the DDR and future replication (Figure 3f). EBV and HIV have both been shown to amplify host centrosomes through genome manipulation, generating chromosomal instability and increasing aneuploidy risk [112,113].

Tumor suppressors are balanced by cyclins [114], which inactivate the tumor suppressors via phosphorylation and advance the cell cycle when checkpoints are satisfied (Figure 3c). For the viral genome to persist in the nucleus, it must evade the p53/p21/pRB cascade and promote cyclin activity (Figure 3d). In fact, p53 was first discovered through its interaction with the polyomavirus transcription factor, LTAg, which binds and inhibits both p53 and pRb [106]. This generated the titular “poly”, meaning “many”, and “oma”, meaning “tumor”, for the family of small dsDNA viruses. Convergent evolution of viral oncogenes has occurred for other neurotropic viruses as well (Table 3). CMV kinase pUL97 directly inactivates pRb, whereas tegument protein pp71 triggers pRb ubiquitination and degradation [108,109]. Furthermore, CMV transcription factor IE2 triggers pRb phosphorylation via cyclin-E1 [107]. EBV latent proteins LMP1 and EBNA3a promote cyclinD1 expression, inactivating pRb through phosphorylation [100,111]. EBNA3c also binds pRb directly, inactivating it. Similar to CMV IE2 targeting of tumor suppressor pRb, EBNA3c promotes ubiquitination and degradation of tumor suppressor p53 [99,110].

### 3.2. Viral Attrition of Telomeres

The protective TTAGGG repeats that cap chromosomes enable lossless replication by DNA polymerases [115]. With each cell cycle, these “telomeres” shorten as DNA polymerase fails to read the terminal ends of linear DNA, indicating cell age in a way similar to rings of a tree trunk. This is performed by Telomerase Reverse Transcriptase (TERT), which extends the telomere through the addition of additional protective repeats, increasing the number of cell cycles a gene can be processed before coding DNA is lost [116]. The shelterin protein complex is also key, as it prevents the IFI16 sensor of the DNA damage response from targeting the uncapped, hanging ends of telomeres, but exogenous toxins (like viruses) can shorten or aberrate telomeres while avoiding the DNA damage response [117]. Innate immune cells exhibit elongated telomeres, an evolutionary selection believed to cope with the telomere expenditure characteristic of infection [118]. Because viruses depend on host cell proliferation for the production of viral progeny, pro-mitotic manipulation by viruses inherently advances telomere attrition and, therefore, advances the speed of cellular aging. Even during AART treatment (a pharmaceutical cocktail of three or more medications targeting HIV entry, expression, and maturation, to generate an imperceptible viral load), telomere length is significantly reduced by latent HIV [119]. Persistent, subclinical viruses like HSV1 and CMV have also been shown to advance the speed of telomere loss through increasing the host’s “pathogen burden” [120,121]. During severe acute infection, as observed with epidemic SARS-CoV-2, severe inflammation has been correlated with rapid, and permanent, telomere attrition [122].

The self-sufficient integration and replication by telomerases mirror viral propagation, suggesting telomeres may have evolved through viral integration into the human genome [123]. Similar to telomerases, DNA viruses avoid recognition by the host DNA-damage response through looping/circularization, terminal capping, and telomerase manipulation [117].

### 3.3. Viral Shift of the Epigenome

Beyond the physical compromise of the host genome, epigenetic modifications that affect the three-dimensional accessibility of genes can be altered by viruses [124,125]. Just as with manipulation of the PQC, DDR, and cell cycle, viruses manipulate the host epigenome to facilitate viral replication at the expense of the host [126]. This accelerates aging and advances mortality [127]. Not all epigenetic alterations can be included in this review (i.e., microRNA, RNA splicing, RNA degradation), but have been reviewed extensively in recent textbooks [128,129]. Basic epigenetic modifications discussed here include inactivating DNA methylation, inactivating histone methylation, and activating histone acetylation.

By promoting DNA methylation at CpG islands, viruses increase obstructive methyl groups, preventing host gene expression and increasing resources for viral gene expression. This resource theft is characteristic of herpesviruses. HSV1 capsid protein, VP26, binds DNA methyltransferase (DNMT), promoting inactivating methylation of host genes [130]. This is a critical function; in the absence of the VP26-DNMT interaction, HSV1 viral copies are significantly reduced. EBV proteins EBNA3a and EBNA3c drive inactivating methylation of the tumor suppressor Bim by selectively activating DNMT activity within the Bim promoter [131]. This methylation is further advanced by another EBV protein, LMP1, which impairs reactivating demethylation by TET [132]. The resultant inhibition of Bim acts in concordance with EBV’s other anti-apoptotic mechanisms discussed above, and DNA hypermethylation has been proposed as a biomarker and therapeutic target for EBV malignancies [133,134]. Other viral infections have been discussed regarding methylation-dependent therapies. CMV exhibits DNA hypermethylation behavior, with CMV seropositivity correlating clinically with a general increase in DNA methylation, epigenetic age, and mortality [135]. Similar hypermethylation patterns have been correlated with other DNA polyomaviruses (i.e., JCV), as well as RNA enteroviruses (i.e., coxsackievirus), coronaviruses, and lentiviruses (i.e., HIV) [136,137,138]. Effective antiviral treatment has shown that this methylation pattern is partially reversible, and correlates with viral load [138,139].

In addition to DNA methylation, viruses alter the acetylation and/or methylation status of histones, inhibiting or promoting coiling into inactive nucleosomes, respectively. The dengue flavivirus capsid directly binds core histones, disrupting host nucleosomes while facilitating viral gene expression [140]. HSV1 and VZV both interact with host protein HCF-1 to promote histone methylation and inactivation of host genes [141]. VZV protein IE63 and EBV protein EBNA3C are both known to recruit deacetylases to the nucleus, reducing accessibility for host gene expression, similar to DNA methylation [142,143]. Conversely, CMV proteins IE1/2 antagonize deacetylases in order to prevent their silencing of the CMV promoter [144]. Therefore, histone deacetylases have been shown to be antiviral in certain infection conditions, and are activated in others, dependent on the virus of interest [145].

## 4. Senescence

Senescence is a cell fate characterized by definitive arrest of the cell cycle, while cells remain alive and display specific secretome and phenotype alterations [146]. Senescence has been demonstrated to cause organism aging and lifespan in vivo [147]. Most cells exhibit a pre-determined lifespan, after which they lose their capacity for proliferation and function [148]. This cycle-dependent decline is termed senescence and varies across cell types based on both intrinsic and extrinsic conditions. Although seemingly in conflict, senescence is a secondary alternative to oncogenesis as it spares the host from lysis, prolonging viral access to the replicative machinery needed for the production of viral progeny. In conjunction with being pro-proliferative, latent viruses exhibit pro-senescence characteristics (Table 4). In addition to inducing senescence of infected cells, viruses are enhanced by systemic senescence—with natural aging of the immune and vascular systems disinhibiting latent or persistent viral infections [149,150].

Immunosenescence is a multifactorial and dynamic phenomenon that affects both innate and acquired immunity during aging [163]. The gradual decline of the immune system is further advanced by chronic diseases. Although the development of this immune phenotype is unclear, it may represent exhaustion of the immune system through several factors, including inflammation, reactive oxygen species, genomic damage, mitochondrial failure, and chronic viral infection [155]. Accumulating evidence shows that several DNA and RNA viruses are inducers of immunosenescence. DNA herpesviruses can upregulate p16, p21, and p53 senescence-associated molecules, induce inflammaging, and metabolic reprogramming of infected cells, replicative senescence, and telomere shortening, as well as epigenetic modification of DNA and histones [164,165]. The same is true for RNA viruses, such as HIV [119].

CMV is the most established neurotropic virus to induce the immune dysfunctions observed in immunosenescence [166]. Although CMV was once considered the leading cause of age-related immune changes in the elderly, recent data are quite contradictory [166,167]. The current opinion is that CMV infection is not directly detrimental, but rather a recurrent stimulation that maintains sustained immunological alertness [168]. This favors a more robust immune response, at the risk of immune exhaustion in chronic infection. There is growing evidence and consensus that chronic antigen stimulation is necessary to cause immunosenescence [151,169]. Infection can lead to the accumulation of terminally differentiated T cells, or memory inflation (Table 4). This brings potential for both premature immune system aging and biological aging.

The immune phenotypes of cells of the immune system are directly or indirectly modified by viral infection, which has implications for immune responses. Previous studies have highlighted the significance of infection as an immune attenuator by affecting the expression of PD1 and CTLA4 [170]. CD4+ and CD8+ T lymphocytes display hallmarks of senescence defined by CD57 expression, which is associated with decreased proliferation capacity and function of these cells [154]. Therefore, it is assumed that the continuous exposure of immune cells to viral antigens during chronic infection drives exhaustion and senescence [171]. This is further supported by CMV-specific T lymphocytes, which show higher levels of proinflammatory cytokines coupled with a lower proliferative response upon stimulation [172]. Although it is difficult to define a singular molecular mechanism, the cycle of inflammation-driven virus reactivation and virus-induced inflammation is clearly implicated in the senescent immune phenotype that has negative consequences for the host.

## 5. Conclusions

In this review, we discuss the overlap of viral pathogenesis with age-related conditions. Using neurotropic viruses as an example, we highlight major mechanisms established to have chronic implications in addition to the more obvious, and symptomatic, dangers of acute infection. Although neurodegenerative conditions (e.g., AD, ALS, FTD, and PD) reflect extremes of stress accumulation, it is critical to note that a spectrum of stressors exists, often subclinically, producing varying speeds at which the brain ages. This is in concordance with intrinsic (i.e., reactive oxygen species produced by metabolism) and extrinsic (i.e., reactive oxygen species produced by physical trauma) stressors and can be countered by protective behaviors (i.e., healthy diet, exercise) [164,173,174]. As described above, neurotropic viruses interact with other stressors, reactivating during deviations from homeostasis due to physical trauma, immune compromise, or systemic aging [4]. Although non-fatal and potentially asymptomatic, the many ways in which persistent viruses age the brain underline the importance of antiviral therapies, vaccines, and preventative care health habits that counter the stresses of unavoidable environmental stresses, like viruses.

## Figures and Tables

**Figure 1 biomolecules-15-00514-f001:**
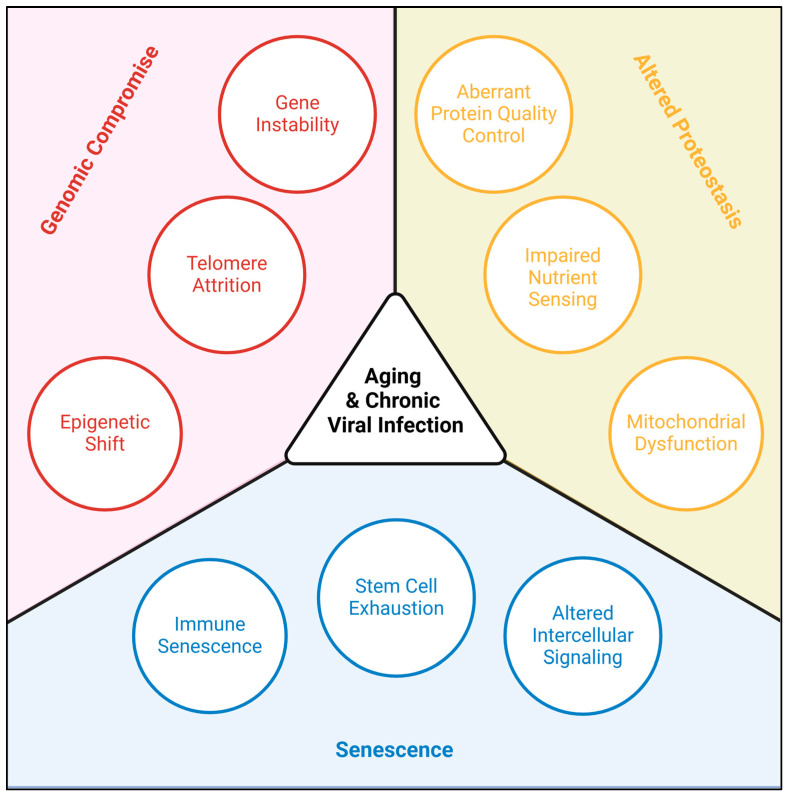
Hallmarks shared by aging and chronic viral infection.

**Figure 2 biomolecules-15-00514-f002:**
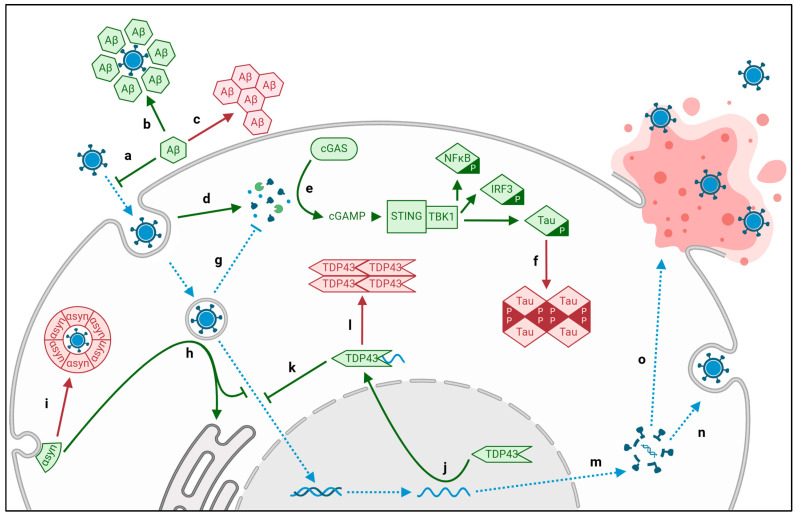
Viral dysregulation of protein quality control. The path of the viral life cycle can be followed by the blue arrows; the healthy response by the cell is noted by green arrows whereas the pathogenic response is noted by red arrows. In the extracellular space, Aβ obstructs viral cell entry (**a**) by forming an obstructive corona on the viral surface (**b**). This can lead to amyloid plaque formation (**c**). Inside the cell, viral molecules are recognized by cGAS (**d**), triggering the STING pathway (**e**), but downstream hyperphosphorylation of tau can produce tau tangles (**f**). Some viruses have evolved anti-cGAS mechanisms, preventing the antiviral cascade (**g**). Alternative antiviral mechanisms by the host include α-synuclein sequestration of virions in the ER (**h**); however, this can lead to formation of Lewy bodies (**i**). Viral RNA within the nucleus translocates TDP43 to the cytoplasm (**j**), where it can obstruct delivery of additional virions to the nucleus (**k**) but can also form obstructive aggregates (**l**). Ultimately, the virus produces viral progeny using host cell machinery (**m**), leading to release and spread via cell-sparing exocytosis (**n**) or fatal lysis (**o**).

**Figure 3 biomolecules-15-00514-f003:**
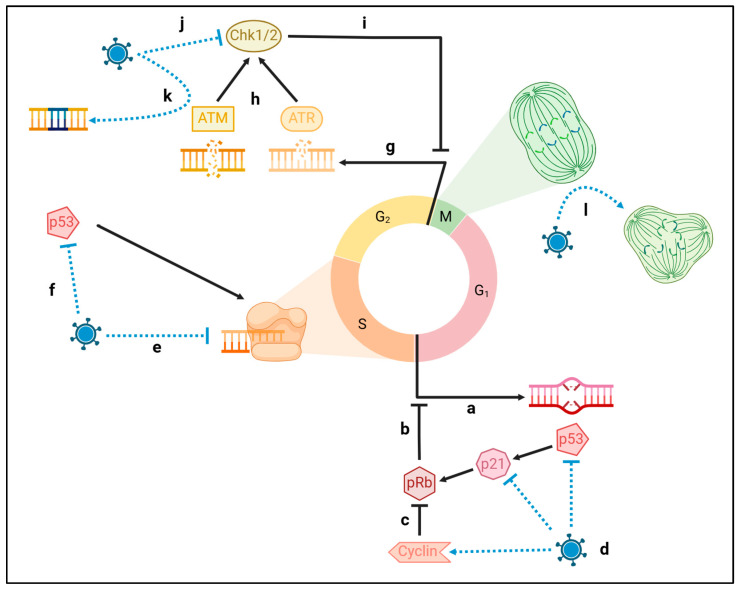
Viral dysregulation of DNA repair responses. The normal cell cycle is annotated by black arrows; viral manipulations are indicated in blue arrows. At the G1S junction, DNA damage triggers tumor suppressor p53, p21, and pRb (**a**), which inhibit progression to the synthesis phase (**b**). Endogenous cyclins balance this process (**c**). Viruses directly impair tumor suppressors p53 and p21 and indirectly promote cyclin inhibition of pRb (**d**), facilitating cell cycle progression regardless of DNA stability. In the synthesis phase, viruses impair telomerases directly (**e**) and indirectly through inhibition of pro-telomerase p53 (**f**). At the G2M junction, DNA damage is identified by ATM and ATR proteins (**g**), activating checkpoint inhibitors Chk1/2 (**h**), which pause the cycle until DNA is repaired (**i**). Viruses directly impair Chk1/2, facilitating progression through the cell cycle (**j**) while also promoting ATM repair of viral genes (**k**). In addition to the aberrant DNA, viruses duplicate centrosomes, leading to asymmetrical division of the already impaired host daughter cells (**l**).

**Table 1 biomolecules-15-00514-t001:** Neurotropic viruses implicated in aging.

Genus	Genome	Capsid	Virus	Associated Diseases
Coronavirus	Linear +ssRNA	Enveloped Icosahedral	SARS-CoV-2	COVID-19
Enteroviruses	Linear +ssRNA	Non-enveloped Icosahedral	Coxsackievirus	Hand-Foot-and-Mouth, Viral Meningitis
			Echovirus	Viral Meningitis
			Poliovirus	Paralytic Poliomyelitis
Flaviviruses	Linear +ssRNA	Enveloped Dimeric αhelix	Dengue	Breakbone Fever
			Japanese Encephalitis	Viral Encephalitis
			West Nile	Viral Encephalitis
Herpesviruses	Linear dsDNA	Enveloped Icosahedral	Herpes Simplex 1	Cold Sores, Viral Encephalitis
			Varicella Zoster	Chicken Pox, Shingles
			Epstein–Barr	Cancer (Lymphoma, Leukemia, Nasopharyngeal Carcinoma), Infectious Mononucleosis, Multiple Sclerosis
			Cytomegalovirus	Congenital Birth Defects, Viral Encephalitis
Polyomaviruses	Circular dsDNA	Non-enveloped Icosahedral	JC	Progressive Multifocal Leukoencephalopathy, Cancer (Glioblastoma, Colorectal Carcinoma)
Lentiviruses	Linear +ssRNA	Enveloped Cone-shaped	Human immunodeficiency virus	Acquired Immunodeficiency Syndrome, HIV-associated Neurocognitive Disorder

+ssRNA represents “positive sense” orientation of an RNA genome, which is capable of immediate translation by host machinery. Further discussion of the human diseases that are associated with these viruses can be viewed in *Fields Virology* [18].

**Table 2 biomolecules-15-00514-t002:** Mechanisms by which neurotropic viruses trigger proteinopathy.

Disease	Proteinopathy	Pathway	Virus	Protein	Citations
Alzheimer’s Dementia	Extracellular Aβ Plaques	Amyloidogenesis	CMV	M45	[23]
			HIV	TAT	[24,25]
			HSVI	gD	[26]
			SARS-CoV-2	S-protein	[27]
	Intracellular Tau Tangles	cGAS-STING	CMV	pUL31 * pUL83 *	[28,29]
			Dengue	NS2B, NS3, NS2B3	[30]
			HIV	GAG *	[31]
			HSV1	ICP27 VP11 *	[32,33]
			SARS-CoV-2	S-protein	[34]
			VZV	ORF9 *	[35]
Amyotrophic Lateral Sclerosis & Frontotemporal Dementia	Cytosolic TDP43 Aggregates	RNA Translocation	CV	2A, 2C	[36,37,38,39,40]
			Echovirus		
			Poliovirus		
			HIV	GAG VIF	[36]
			HERV-K	ASRGL1 *	[41]
			SARS-CoV-2	S-protein	[27]
Parkinson’s Disease & Lewy Body Dementia	Lewy body Aggregates	Endoplasmic Reticulum Sequestration	CMV	Envelope	[42]
			EBV		[43,44]
			Dengue		[45]
			HIV		[46,47]
			JEV		[48]
			WNV		[44]
			SARS-CoV-2	N-protein S-protein	[49,50]

* These proteins physically obstruct cGAS access to viral nucleic acids, preventing appropriate activation of the antiviral cascade.

**Table 3 biomolecules-15-00514-t003:** Mechanisms by which neurotropic viruses promote cell cycle progression.

Target	Virus	Protein	Mechanism	Citations
ATM/ATR	EBV	EBNA3c	Evasion of ATM via p53 degradation	[99]
		LMP1	Transcriptional downregulation of ATM	[100]
	HIV	VPR	Chromatin binding activates ATR	[96]
Chk1/2	EBV	EBNA3a	Inactivation by direct binding	[101]
	HIV	VPR	Inactivation by phosphorylation	[102]
	SARS-CoV-2	ORF6 NSP13	Proteolysis Autophagy-mediated degradation	[98]
p53	EBV	EBNA3c	Ubiquitin-directed degradation	[103]
	JCV	LTAg	Inactivation by direct binding	[104,105,106]
pRb	CMV	IE2	Inactivating phosphorylation via Cyclin-E1	[107]
		pp71	Ubiquitin-directed degradation	[108]
		pUL97	Inactivation by phosphorylation	[109]
	EBV	EBNA3	Inactivation by direct binding Inactivating phosphorylation via Cyclin-D1	[110]
		LMP1	Inactivating phosphorylation via Cyclin-D1	[111]
	JCV	LTAg	Inactivation by direct binding	[104,105,106]

**Table 4 biomolecules-15-00514-t004:** Adaptive immune changes associated with chronic viral infection.

Virus	Immune Changes	Citations
CMV	Decrease in percentage of CD16− NK cells	[151]
	Decrease in percentage of CD16+/CD56bright NK cells	
	Increased CD16+/CD56− subset	[152]
	Increased CD8+ T-cells with high CD244 expression	[153]
	Increased CD4+ and CD8+ effector memory cells	[154]
	Exhaustion of peripheral T-cell compartments	[155]
	Accumulation of terminally differentiated, apoptosis-resistant, CMV-specific CD8+ lymphocytes	[156,157]
	Reduced diversity of TCR repertoire	[158]
EBV	Increase in differentiated phenotype markers (i.e., KLRG1)	[159]
	Increase in terminally differentiated T-cells	
	Reduced diversity of TCR repertoire	
VZV	Increased population of CD57+, terminally differentiated NK cells	[160]
	Impaired Type I IFN pathway	[161]
	Impaired production of pro-inflammatory cytokines	
	Reduced frequency of VZV-specific memory T cells	[162]

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
