# Peer review of "Neurotropic Viruses as Acute and Insidious Drivers of Aging"

_biomolecules, 2025, doi:10.3390/biom15040514_

Round 1
Reviewer 1 Report
Comments and Suggestions for Authors
This review is focused on the impact of viral pathogenesis on aging and discuss how active and latent viruses contribute to aging. They emphasize that viruses might contribute to premature aging via viral induction of common stress response pathways. Examples include: dysregulated homeostasis by exogenous viral proteins and overwhelmed protein quality control mechanisms, DNA damage through direct integration and epigenetic manipulation, immune-mediated oxidative stress and immune exhaustion, and general energy theft that is amplified in an aging system. Surprisingly, the authors also highlights the long-term importance of vaccines and antivirals in addition to their acute benefits. I suggest to mention that viruses have been prosed as a tool to redress the astrocyte vs neurones balance in the ageing brain (see,doi: 10.1007/s11357-019-00084-0; doi: 10.3389/fnagi.2019.00334;doi: 10.3390/biom14030289; doi: 10.4103/1673-5374.306064)
Comments on the Quality of English LanguageEnglish is good to convey the scientific content of the manuscript
Author Response
- Comment 1: “I suggest to mention that viruses have been proposed as a tool to redress the astrocyte vs neuron balance in the aging brain.”
- Response 1: We agree with this comment and appreciate the effort on the part of the reviewer to suggest specific, additional citations valuable to this text; therefore, we have incorporated lines 41-48: “The neuro-penetrance of these viruses is so extensive, in fact, they have been proposed, extensively, as a non-invasive vector for therapeutic delivery to the brain [5,6]. Specific to aging, viral vectors have been proposed to deliver neurotrophic factors for the treatment of PD [7]; genetic interventions post-ischemic stroke [8,9]; GABA antagonists for the treatment of cognitive decline [10]; and epigenetic modulators for intervention of epigenetic aging [11].”

Reviewer 2 Report
Comments and Suggestions for Authors
This manuscript provides an excellent and original review of the similitudes and interactions between viral infections and aging. It is remarkably written, strongly documented by pertinent and updated references, and it will be helpful to both basic and medical researchers. However it deserves some minor improvements before being ready to publish, as follows.
Scientific issues
Lines 48-54 : Please authors add a concise definition of “positive sense” viruses; either between bracketts after the first one of the list, or as a legend for Table 1.
L. 160: Add a short explicitation of the ubiquitin-proteasome pathway of PQC, after “by proteolysis”.
L. 198: Please provide here concise explanation of the ATM and ATR pathways for single/double DNA breaks (add one sentence, and one more reference to a basic pertinent review).
L. 225: Add “(see section 3.3 for functional description)” after “the telomeric repeats”.
L. 267: Add one basic review reference for cyclin balancing of tumor suppressors.
L.289: In order to complete the excellent overview of the telomere structure and function in lines 282-289, please authors add one more sentence to introduced the intrinsic telomerase and its role.
L.291: Provide concise presentation of AART.
L. 343-344: Rewrite the sentence into “Senescence is a cell fate characterized as a definitive arrest of cell cycle while cells remain alive and display specific alterations of secretome and phenotype (Gorgoulis et al, 2019, Cell 179(4):813-27). Senescence in vivo has been demonstrated to cause organism aging and lifespan (Baker DJ et al, 2016, Nature 530:184-9).”
Author Response
- Comment 1: “L. 48-54: Please authors add a concise definition of “positive sense” viruses; either between brackets after the first one of the list, or as a legend for Table 1.”
- Response 1: We agree with this necessary explanation, which has been added as a footnote in Table 1 on page 3: “+ssRNA represents “positive sense” orientation of an RNA genome, which is capable of immediate translation by host machinery.”
- Comment 2: “L. 160: Add a short explanation of the ubiquitin-proteasome pathway of PQC after “by proteolysis.”
- Response 2: We are, once again, in agreement with this necessary explanation, which has been added in addition to a brief explanation of the autophagy lysosomal pathway on page 6, lines 146-150: “In the former mechanism, basal ubiquitination of substrates directs aberrant or foreign proteins to be degraded in proteosome complex; amino acids are then recycled into other polypeptides [69]. In the latter mechanism, a stress-induced chaperone identifies a substrate, engulfs it in a lysosome, and directs its lysosomal degradation through acidic pH and proteases [70].”
- Comment 3: “L. 198: Please provide here a concise explanation of the ATM and ATR pathways for single/double DNA breaks (add one sentence, and one more reference to a basic pertinent review).”
- Response 3: We appreciate the detailed request provided by this reviewer and have added the requested description and citation on page 7 lines 188-190: “At the G2M junction, ATM and ATR identify double and single stranded breaks, respectively [84,85]. ATM activates p53 and Chk, ATR activates Chk2, and altogether these DDR enactors halt cell cycle progression to mitosis.”
- Comment 4: “L. 225: Add (see section 3.3 for functional description) after the telomeric repeats.”
- Response 4: This helpful connection between sections was incorporated on page 8 line 227: “This includes the telomeric repeats (see section 3.3 for functional description), resulting in TERT-dependent upregulation of the cell cycle.”
- Comment 5: “L. 267: Add one basic review reference for cyclin balancing of tumor suppressors.”
- Response 5: To improve the validity of this sentence we have added citation 114 to page 9 line 258 as requested.
- Comment 6: “L. 289: In order to complete the excellent overview of the telomere structure and function in lines 282-289, please authors add one more sentence to introduce the intrinsic telomerase and its role.”
- Response 6: To conform with this apt recommendation for a more thorough illustration of the telomerase complex, we added a more detailed explanation on page 10 lines 274-288: “This is performed by Telomerase Reverse Transcriptase (TERT), which extends the telo-mere through the addition of additional protective repeats, increasing the number of cell cycles a gene can be processed before coding DNA is lost [116].”
- Comment 7: “L. 291: Provide concise presentation of AART.”
- Response 7: A short description of AART therapy was added as a parenthetical on page 10 lines 286-288: “a pharmaceutical cocktail of three or more medications targeting HIV entry, expression, and maturation, to generate imperceptible viral load”
- Comment 8: “L. 343-344: Rewrite the sentence into ‘Senescence is a cell fate characterized as a definitive arrest of cell cycle while cells remain alive and display specific alteration of secretome and phenotype. Senescence in vivo has been demonstrated to cause organism aging and lifespan’.”
- Response 8: We appreciate this amendment and have incorporated the change and accompanying citations on page 11 lines 339-341.

Reviewer 3 Report
Comments and Suggestions for Authors
This review addresses the role of viral infections in neurodegenerative diseases.
- Critique Tables 1, 2 and 3 do not contain any references, so it is not possible to check the statements.References must be inserted here.
- The viruses listed interact with the DNA damage response and can, it is explained, cause breaks in telomeres.I miss a clear statement about which of the viruses are clastogenic, i.e. cause chromosomal aberrations (as shown years ago for SV40)?
- Is there clear evidence that Parkinson's and Alzheimer's disease are caused by viral infections?Please provide a clear, unambiguous statement with references.
- Is there evidence that glioblastomas and colorectal cancer are caused by viruses, as Table 1 suggests?Please provide a clear statement with references here too.
Author Response
- Comment 1: “Tables 1, 2 and 3 do not contain any references, so it is not possible to check the statements. References must be inserted here.”
- Response 1: We wholeheartedly agree with this comment and have added the necessary citations to Table 1 (page 3, footnote), Table 2 (page 4, column 6), and Table 3 (page 9, column 5). Missing citations were also identified and addressed in Table 4 (page 11, column 3).
- Comment 2: “The viruses listed interact with the DNA damage response and can, it is explained, cause breaks in telomeres. I miss a clear statement about which of the viruses are clastogenic, i.e. cause chromosomal aberrations (as shown years ago for SV40)?”
- Response 2: We recognize the flaw in excluding “clastogenic” as a descriptor in Section 3. Genomic Compromise. We have incorporated more explicit language (Page 7, line 199) with citations for HIV [88-91], EBV [95], and CMV [92] in the accompanying paragraphs.
- Comment 3: “Is there clear evidence that Parkinson's and Alzheimer's disease are caused by viral infections? Please provide a clear, unambiguous statement with references?”
- Response 3: A causal relationship between viral infection, PD and AD cannot be confirmed within the constraints of retrospective analysis, however, the publications that provide evidence in favor of this relationship are detailed in Table 2 (page 4) with updated Citations (column 6).
- Comment 4: “Is there evidence that glioblastomas and colorectal cancer are caused by viruses, as Table 1 suggests? Please provide a clear statement with references here too.”
- Response 4: The overview table “Table 1: Neurotropic viruses implicated in aging” draws from the most recent edition of Fields Virology, which recognizes the increased risk of cancer following polyomavirus infection. We have added that major, shared citation to the footnote of Table 1 on page 3. The proposed mechanisms of respective viral oncogenesis are discussed in citations in the new Table 3 Column 5.

Round 2
Reviewer 1 Report
Comments and Suggestions for Authors
The authors have successfully addressed my observations. The manuscript can be published in its present form.
Reviewer 3 Report
Comments and Suggestions for Authors
The revised version is clearly improved.